# Study on Microstructural Evolution and Mechanical Properties of Mg-3Sn-1Mn-xLa Alloy by Backward Extrusion

**DOI:** 10.3390/ma16134588

**Published:** 2023-06-25

**Authors:** Xuefei Zhang, Baoyi Du, Yuejie Cao

**Affiliations:** 1School of Mechanical Engineering, Shenyang University, Shenyang 110044, China; 2School of Aeronautics, Chongqing Jiaotong University, Chongqing 400074, China

**Keywords:** Mg alloy, backward extrusion, tensile strength, failure mechanism

## Abstract

Mg-3Sn-1Mn-xLa alloy bars were prepared using backward extrusion, and the effects of the La content on the microstructures and mechanical properties of the alloy were systematically studied using an optical microscope (OM), X-ray diffraction (XRD), scanning electron microscope (SEM), transmission electron microscope (TEM), and tensile tests. The results of this research show that the Mg_2_Sn phases were mainly formed at the α-Mg grain boundaries and within the grains in the Mg-3Sn-1Mn alloy. After adding a certain amount of La, the plate-shaped MgSnLa compounds consisting of Mg_17_La_2_, Mg_2_Sn, and La_5_Sn_3_ gradually disappeared in the α-Mg matrix and grain boundaries. With an increase in La content, the Mg_2_Sn phase in the crystal was gradually refined and spheroidized. When the content of La reached 1.5%, the tensile strength of the alloy reached 300 Mpa and the elongation reached 12.6%, i.e., 25% and 85% increases, respectively, compared to the Mg-3Sn-1Mn alloy. The plate-shaped compound of Mg-3Sn-1Mn-1.5La had an average length of 3000 ± 50 nm, while the width was 350 ± 10 nm. Meanwhile, the extruded alloy’s grain size was significantly refined, and there were many small cleavage steps and dimples in the fracture surface of the alloy. When the La content reached 2%, the alloy performance showed a downward trend due to the coarsening of the grains. The formed plate-shaped MgSnLa compounds and Mg_2_Sn phases were consistent with the α-Mg matrix. They effectively pinned the dislocations and grain boundaries, which is the main reason for strengthening the mechanical properties of extrusion alloys.

## 1. Introduction

Magnesium and its alloys have the advantages of low density, high specific strength, and stiffness and are widely used in aerospace, national defense, the military industry, the automotive industry, and other fields. However, the low plasticity at room temperature, poor heat, and poor corrosion resistance of traditional magnesium alloys limits their wide application [1,2,3,4,5]. Alloying is an effective method to improve the mechanical properties of magnesium alloys [6]. Previous studies have shown that, when Sn is added to magnesium, the Mg_2_Sn phase formed has a high melting point and high hardness, which can effectively improve the mechanical properties and thermal stability of the alloy, as well as improve the high-temperature strength and creep resistance of magnesium alloys [7,8,9,10,11,12]. In addition, rare earth strengthening is also one of the methods that can effectively improve the mechanical properties of magnesium alloys, because adding rare earth can improve the deformation ability and strength of magnesium alloys through the precipitation hardening and solid solution strengthening mechanisms [13,14,15]. Current research results indicate that adding rare earth elements such as La, Ce, Ga, Y, and Nd to magnesium alloys can effectively refine the α-Mg grains and improve the microstructure of the alloy, thereby significantly improving the comprehensive mechanical properties of the magnesium alloys [16,17,18,19]. Zengin et al. [18] studied the mechanism of the effect of La on the grain refinement process of magnesium alloys. They found that, during the solidification process of the alloy, the enrichment of La atoms at the solid–liquid interface accelerated the nucleation rate while inhibiting the growth of dendrites.

Grain refinement is a critical way to improve the plasticity and toughness of magnesium alloys [5]. In addition to adding rare earths, high-temperature thermoplastic deformation is also a very effective method. High-temperature large plastic deformation can not only effectively eliminate internal defects and voids in as-cast magnesium alloys but also plays a significant role in grain refinement, improving the alloy microstructure’s uniformity and consistency [20]. The grain refinement after extrusion is attributed to typical dynamic recrystallization (DRX) occurring during thermoplastic deformation [21,22]. During plastic deformation, high-density dislocations are introduced into coarse grains, which aggregate to promote the formation of low-angle grain boundaries (LAGBs). As the strain increases, a transition occurs from small-angle grain boundaries to high-angle grain boundaries (HAGBs), resulting in dynamic recrystallization (DRX). During dynamic recrystallization, precipitates can pin dislocations and grain boundaries, inhibiting grain growth [11,23]. Some studies have reported that, on the one hand, precipitated phases bind dislocations, increasing the driving force for recrystallization and promoting dynamic recrystallization (DRX). On the other hand, precipitates bind low-angle grain boundaries (LAGBs) and grain boundaries generated by dynamic crystallization (DRX), inhibiting the transition from low-angle grain boundaries to high-angle grain boundaries (HAGBs), thereby delaying dynamic recrystallization [24,25,26,27]. Furthermore, other studies have also found that a series of alloys that have undergone alloying and rare earth alloying, such as Mg-Sn-Ca-Mn [13,28,29], Mg-Sn-Mn [30,31], Mg-Sn-Zn-Mn [32,33,34], and Mg-Gd-Y-Zn [35,36], exhibit excellent mechanical properties at high temperatures and at room temperature.

Magnesium alloys with a hexagonal close-packed (HCP) crystal structure are difficult to process at room temperature. Currently, they are mainly prepared by casting and thermoforming processes. Based on the above research, Zhao et al. [11,37,38] designed a Mg-3Sn-1Mn-1La alloy using a continuous rheo-rolling method. The Mg_2_Sn phase mainly exists in the Mg-3Sn-1Mn alloy without adding La. After adding a certain amount of La, MgSnLa metal compounds composed of the La_5_Sn_3_, Mg_2_Sn, and Mg_17_La_2_ phases are generated. The presence of these metal compounds improves the room-temperature and high-temperature properties of magnesium alloys. Magnesium alloys can also achieve grain refinement through hot-extrusion deformation and precipitation, which positively impact the alloy’s microstructure and comprehensive mechanical properties [18,26]. However, reports have yet to be made on preparing rare earth magnesium alloys by backward extrusion and their strengthening effects on the microstructure and properties.

In this study, the effects of La on the microstructure and properties of hot-formed Mg-3Sn-1Mn-xLa alloy wire produced by the backward-extrusion process were studied. The objective of the study described here is to reveal the effect of the precipitates on grain refinement during the hot-forming process and to elucidate the effect of the rare earth phase on the mechanical properties of the Mg-3Sn-1Mn-xLa alloy. The results and related discussions will provide an important basis for understanding the fine-grain strengthening mechanism of Mg-3Sn-1Mn alloys and developing high-performance La-containing magnesium alloys.

## 2. Experimental Procedure

Mg-3Sn-1Mn-xLa alloys were prepared by melting pure Mg (99.99% purity), pure tin (99.9% purity), Mg-75Mn (wt.%), and Mg-30La (wt.%). Pure Mg and all other raw materials were preheated in the oven to eliminate water vapor. A resistance furnace (3 kW, SG2-3-9, Shenyang General Furnace Manufacturing Co., Ltd., Shenyang, China) was used to melt Mg alloys. The magnesium ingots were then melted in a clay graphite crucible using an electric resistance furnace at 700 °C to ensure complete homogenization, and the dried Sn, Mn, and Mg-La master alloys were added in sequence. After mechanical stirring, the gas and slag were removed. The melts were then degassed with argon and poured into a low-carbon steel mold of 180 mm height and 40 mm diameter. The chemical composition of the Mg alloy is shown in Table 1.

The backward-extrusion experiment in this paper used a 300 T single-column vertical hydraulic compressor to prepare extruded samples. The schematic diagram of its working principle is shown in Figure 1. Before extrusion processing, the magnesium alloy as-cast samples were homogenized at 390 °C for 6 h [29,34], while the extrusion cylinder was preheated to 350 °C. Afterward, backward extrusion was carried out at 380 °C to adjust the bar diameter from 40 mm to 10 mm, with an extrusion ratio of 16.

The mechanical properties of the extruded specimens were tested on a SANS electronic universal tensile testing machine. The geometric dimension of the tensile specimens was designed according to the GB/T 16865-2013 standard [39], and the detailed sample size is shown in Figure 2. The phase identification of the surface was performed using X-ray diffraction (XRD). The specimens were polished and etched with 15 mL HCl + 56 mL C_2_H_5_OH + 47 mL H_2_O. The microstructure observation and microanalysis were performed using an OLYMPUS DSX500 optical microscope and a Hitachi S-4800 II scanning electron microscope. The TEM observations were performed using a field-emission-gun (FEG) Tecnai G^2^ 20 microscope operating at an accelerating voltage of 200 kV.

## 3. Results and Discussions

### 3.1. Microstructure of As-Cast Alloys

Figure 3 shows the as-cast metallographic microstructure of Mg-3Sn-1Mn-xLa alloys, mainly composed of primary α-Mg and secondary phases. The α-Mg in the alloy specifically presented a dendritic structure, and a large number of grayish-white near-dendritic forms were α-Mg matrix. At the same time, the secondary phase was black and mainly distributed along the grain boundaries. The as-cast structure of the alloy without the addition of La was primarily composed of coarse α-Mg dendrites, interdendrites, and intermetallic compounds between the dendritic arms. The average grain size of the alloy was 150 ± 5 µm; its distribution is shown in Figure 3a. After the addition of La, a large number of black intermetallic compounds appeared along the grain boundaries of the alloy (Figure 3b). With the increase in La content, the number of eutectic compounds increased gradually, and the dendrites started to show refinement (Figure 3c), which was mainly because the second-phase particles formed during solidification prevented the further growth of the dendrites, thus developing finer dendrites. When the La content reached 1.5%, the refinement of the alloy’s dendrites was most obvious, and the second phase near the grain boundary increased significantly. The average size of the grains showed a gradual increase to 50 ± 2 µm. The distribution was relatively uniform, which is due to the enrichment of La atoms at the solid–liquid interface during solidification leading to the acceleration of the nucleation rate and the restriction of dendrite growth (Figure 3d). When the rare earth content continued to increase, the excessive enrichment of the precipitated phases increased the segregation, and the grains appeared coarsened instead. The average grain size of the alloy increased to 100 ± 3 µm, as shown in Figure 3e. A more detailed display and analysis of the precipitates will be further introduced in the following sections.

Figure 4 shows the effect of La concentration on precipitate formation in the Mg-3Sn-1Mn-xLa (wt.%) alloy as studied by X-ray diffraction for different La concentrations. It indicates that the sample was mainly composed of α-Mg matrix, and the second-phase Mg_2_Sn was only detected in the Mg-3Sn-1Mn alloy (curve (1)). However, with the addition of La (curve (2)), the diffraction peaks of the Mg_2_Sn phase began to weaken, indicating that with the increase in La concentration, Sn preferentially formed new phases with La, resulting in a decrease in the Mg_2_Sn phases. As shown in curve (4) in Figure 4, when the La concentration increased to 1.5 wt.%, the diffraction peak intensities associated with the Mg_2_Sn phase were further decreased. Furthermore, when the La concentration was 0.5 wt.%, the diffraction peaks of the La_5_Sn_3_ phase and Mg_17_La_2_ phase appeared. As the La concentration increased to 1.5 wt.%, the diffraction peak of the La_5_Sn_3_ phase increased and began to appear at a new peak position. At the same time, the diffraction peaks of the Mg_17_La_2_ phase also appeared at a new position, indicating a further increase in the Mg_17_La_2_ phase. Generally speaking, the difference in the diffraction peak intensity of Mg-3Sn-1Mn-xLa manifests that the amount of La_5_Sn_3_ and Mg_17_La_2_ precipitates increased apparently with increasing La content.

Figure 5 shows the as-cast SEM microstructure and EDS spectrum analysis of Mg-3Sn-1Mn and Mg-3Sn-1Mn-1.5La alloys. It can be seen from Figure 5a,c that there was an irregularly distributed second phase near the grain boundary in the Mg-3Sn-1Mn alloy. Combined with the XRD phase analysis in Figure 4, the second phase was mainly the Mg_2_Sn phase. In the Mg-3Sn-1Mn-1.5La alloy, a large number of plate-shaped compounds were distributed at the grain boundaries, and a small number of small, elongated phases were also present in the grains. Through the XRD phase analysis results in Figure 4, it can be seen that the plate-shaped compounds comprised a ternary mixture of La_5_Sn_3_, Mg_2_Sn, and Mg_17_La_2_ phases.

### 3.2. Microstructure Morphology of Backward-Extrusion Alloy

The microstructure morphology of Mg-3Sn-1Mn and Mg-3Sn-1Mn-1.5La alloys after backward extrusion in the transverse direction (TD) are shown in Figure 6a,b. Compared with the microstructure of the as-cast Mg alloy in Figure 4, the microstructure had undergone significant changes, with substantial grain refinement and many precipitates appearing on the Mg matrix. The main reason is that Mg alloys are prone to dynamic recrystallization during the hot-forming process to form new recrystallized grains. The accumulation of dislocations at the grain boundaries or second-phase particles of dynamic recrystallization grains can promote the occurrence of dynamic recrystallization [23,40,41]. Furthermore, the microstructure morphology of Mg-3Sn-1Mn and Mg-3Sn-1Mn-1.5La alloys in the extrusion direction (ED) is shown in Figure 6c,d. As can be seen from Figure 6c, the grain structure of the alloy was significantly refined after backward extrusion and typical refined equiaxed grains appeared. This is mainly attributed to the dynamic recrystallization and heterogeneous nucleation of recrystallized grains of the Mg_2_Sn phase precipitated during the extrusion process and the hindering effect on grain growth. Previous studies have shown that equiaxed grains with a uniform distribution can hinder the formation of the void, thus increasing the elongation to tensile failure [1]. Compared to Mg-3Sn-1Mn alloy, the Mg-3Sn-1Mn-1.5La alloy exhibited many black fibrous structures with streamline distribution along the extrusion direction, which probably resulted from the extrusion deformation of precipitates.

The SEM microstructures of the Mg-3Sn-1Mn and Mg-3Sn-1Mn-1.5La alloys in the backward-extrusion state are shown in Figure 7a,b. By comparison, it was found that more dense precipitates appeared on the matrix surface of the Mg-3Sn-1Mn-1.5La alloy. Combined with previous XRD analysis, it was found that the precipitates were mainly composed of a mixture of Mg_2_Sn and MgSnLa. In addition, it can be observed from Figure 7c,d that the grains underwent apparent fragmentation during the extrusion deformation process, and the second phase was mainly distributed densely along the grain boundaries. From this, it can be seen that the distribution of the second phase was within the Mg crystal and at the grain boundaries. To further observe and analyze the structure and composition of the precipitates, Figure 8 shows the TEM microstructure and EDS energy spectrum analysis of the two alloys with the extrusion state. It can be seen from Figure 8a,c that spherical secondary precipitates were evenly distributed in the crystals of the two alloys. The selected area’s electron diffraction pattern showed a single phase, and the EDS energy spectrum analysis was carried out at points A and B, respectively, as shown in Figure 8b,d. It can be determined as the Mg_2_Sn phase and the Mg_2_Sn phase in the crystal of the Mg-3Sn-1Mn-1.5La alloy was more refined and spheroidized than that of the Mg-3Sn-1Mn alloy. Zhao et al. [11,31,37] indicate that the TEM images and superimposed diffraction patterns of the matrix show that the Mg_2_Sn phase was short-rod, lath-shaped, and spherical with hundreds of nanometers in thickness and diameter in the Mg-3Sn-1Mn alloy. However, the plate-shaped MgSnLa compound can be observed in the as-prepared Mg-3Sn-1Mn-1La alloy. Figure 9a–d shows the TEM structure morphology and EDS energy spectrum analysis of the Mg-3Sn-1Mn+1.5La alloy’s grain boundary location. It can be seen that a large number of blocky and plate-shaped second phases were distributed at the grain boundary. The selected area’s electron diffraction pattern showed that it was a symbiotic mixed phase. The EDS analysis results at the junction points A and B are shown in Figure 9b,d. According to the XRD phase analysis, the blocky or plate-shaped compounds were mainly a MgSnLa mixture composed of the La_5_Sn_3_, Mg_2_Sn, and Mg_17_La_2_ phases. The spherical phase with smaller size (average diameter ≤ 40 nm) was the Mg_2_Sn phase.

### 3.3. Tensile Mechanical Properties

Figure 10 shows the stress–strain response curve of Mg-3Sn-1Mn-xLa backward-extrusion alloy under a room-temperature tensile test. It can be seen from the figure that the stiffness of all specimens remained consistent during the initial linear elasticity stage. The ultimate tensile strength of the Mg-3Sn-1Mn alloy was 240 Mpa, and the elongation at tensile failure was 6.8%. After adding 0.5% La, the mechanical properties of the alloy underwent significant changes, with a substantial increase in ultimate tensile strength of 270 Mpa and an elongation of 9.2%. The effect of rare earth strengthening on mechanical properties was very obvious. As the La content continued to increase to 1%, the tensile strength of the alloy also further increased to 285 Mpa, with an elongation of 10.4% at this time. When the La content increased to 1.5%, the tensile strength of the alloy also reached 300 Mpa, and the elongation at this time was as high as 12.6%. When the La content continued to increase to 2%, the tensile strength of the alloy was 294 Mpa, and the elongation was 10.7%, indicating a specific decrease in performance. This decrease was probably caused by the excessive La, reducing the uniformity and refinement of the metal microstructure. From the above phenomena, we can conclude that adding La can effectively improve the mechanical properties of the Mg-3Sn-1Mn alloy. As the La content increases, the mechanical properties of the alloy also improve. However, when the La content reaches 2%, the alloy performance begins to decline, indicating that the addition of La should be controlled within a reasonable range. Compared with the Mg-3Sn-1Mn alloy without adding the La element, the tensile strength and elongation of the Mg-3Sn-1Mn-xLa alloy with a 1.5% content increased by 25% and 85%, respectively. This improvement has significance for the performance strengthening and optimization of magnesium alloys.

Figure 11 shows the SEM fractography of the tensile-tested samples from backward-extruded parts at room temperature. It can be seen from Figure 11a that the fracture surface of the Mg-3Sn-1Mn alloy was mainly composed of rough tearing edges belonging to a typical brittle fracture. With the increase in La content, the fracture morphology of the alloy was significantly refined, and the grain distribution tended to be uniform, possessing the apparent refinement of an equiaxed grain fracture. At the same time, some small ductile dimples also appeared, with a mixed fracture feature of brittleness and toughness, as seen in Figure 11b. This was also the direct reason for the significant improvement in tensile strength and elongation with the addition of La seen in Figure 10. Continuing to increase the La content, the number of dimples on the alloy cross-section began to grow, and the tearing edges significantly decreased and became smaller. The fracture surface of the alloy began to appear as a cleavage layer, and the plastic characteristics further improved, with the typical characteristics of brittle fracture and cleavage fracture. The number of fracture dimples further increased, as shown in Figure 11c. When the La content continued to increase to 1.5%, the morphology of the fracture pits increased significantly, and the pits’ depth also increased significantly. The cleavage phenomenon began to decrease; the tearing edges disappeared; and the slight dimples were tightly arranged. The grain refinement and homogenization phenomenon was more prominent, as shown in Figure 11d. However, when the La content reached 2.0 wt.%, the distribution of the fracture toughness pits began to coarsen (Figure 11e), which also confirms why the performance (Figure 10) began to decline.

### 3.4. Strengthening Mechanisms

In the Mg-3Sn-1Mn-xLa alloy, the Mn element is wholly dissolved in the α-Mg matrix and plays a solid solution strengthening role [33,42,43]. The size and distribution of the Mg_2_Sn and Sn_3_RE_5_ phases have a significant impact on the mechanical properties of alloys [9,11,44,45]. Sn_3_RE_5_ is a cracked, rod-shaped phase formed by the combination of Sn and rare earth elements in Mg alloys. The Sn_3_RE_5_ phases exhibit superior high-temperature stability. The interface between the Mg_2_Sn and Sn_3_RE_5_ phase and the α-Mg matrix is relatively stable, and it is difficult to nucleate and generate microcracks during deformation. Moreover, the formation of the Mg_2_Sn phase can effectively hinder dislocation slip and improve the strength of the alloy. Figure 12 shows the TEM microstructure of the Mg-3Sn-1Mn-xLa alloy sample after tensile deformation. It can be seen from Figure 12a,b that Mg_2_Sn in the Mg-3Sn-1Mn alloy had a pinning effect on dislocations, which can effectively hinder the slip of dislocations. When dislocations form and move between Mg_2_Sn phases, due to the pinning effect of Mg_2_Sn relative to dislocations, dislocations aggregate between the Mg_2_Sn phase particles, which hinders the further diffusion of dislocations and improves the strength of the alloy. When the content of the La element reaches 1.5 wt.%, as seen in Figure 12c,d, the plate-shaped MgSnLa compounds formed near the grain boundary grow perpendicular to the grain boundary direction and extend into the grain, pinning the grain boundary to prevent the grain boundary sliding. At the same time, the plate-shaped MgSnLa compounds can also pin dislocations to prevent dislocation sliding. When the dislocations move to the plate-shaped compound, due to the blocking effect of the plate-shaped MgSnLa compounds on the dislocations, the dislocations accumulate around it. When dislocations move between the plate-shaped compounds, they are surrounded by plate-shaped MgSnLa compounds, which hinder their further movement, thus improving the strength of the Mg alloy. Furthermore, from Figure 12e, it is shown that the hindrance of Mg_2_Sn relative dislocations within the crystal is evident. The interaction between dislocations and Mg_2_Sn exhibits a bypass mechanism, resulting in Orowan strengthening [31]. Some studies have also shown that when dislocations encounter the Mg_2_Sn phase, they bypass the phase to undergo Orowan strengthening, further improving the strength of the alloy [46,47]. As the content of the La element increases, the Mg_2_Sn phase gradually refines and spheroidizes, and its distribution tends to be uniform. The pinning effect of Mg_2_Sn relative to dislocations and its hindrance to dislocation movement is enhanced, strengthening the alloy. When the content of the La element reaches 2.0 wt.%, the plate-shaped MgSnLa compounds gradually increase, which makes the grain boundary coarser and more pronounced. The pinning effect of the plate-shaped compounds on the grain boundary is weakened, so the mechanical properties of the alloy are reduced.

## 4. Conclusions

This study investigated the microstructure evolution and mechanical properties of Mg-3Sn-1Mn-xLa (x = 0, 0.5, 1.5, 2.0 wt.%) alloys prepared using the backward-extrusion method. Based on the OM, XRD, SEM, and TEM observations and tensile properties, the main research findings are as follows:In the cast and extruded Mg-3Sn-1Mn alloys, the main distribution was α-Mg grains and Mg_2_Sn phases located at grain boundaries and within grains. Mg_2_Sn phases can improve alloy properties by pinning dislocations and hindering dislocation slip.With the addition of La, the plate-shaped MgSnLa compounds composed of Mg_17_La_2_, Mg_2_Sn, and La_5_Sn_3_ phases began to be densely distributed along the α-Mg grain boundary, which can act as a pinning for grain boundaries and hinder dislocation slip. At the same time, the Mg_2_Sn phases exhibited significant refinement and spheroidization.With the increase in La, the mechanical properties of the extruded Mg-3Sn-1Mn-xLa alloy were significantly improved. When the La content reached 1.5%, the tensile strength at room temperature increased to 300 Mpa, and the elongation reached 12.6%, i.e., 25% and 85% increases on the tensile strength (240 MPa) and elongation (6.8%) of the Mg-3Sn-1Mn alloy without La added, respectively. In addition, the grain size of rare earth magnesium alloy was significantly refined, and there were numerous small cleavage steps and dimples in the fracture surface of the sample. However, when the La content continued to increase to 2%, the alloy’s properties began to show a downward trend, and there was a coarsening phenomenon in the distribution of the fracture surface.

## Figures and Tables

**Figure 1 materials-16-04588-f001:**
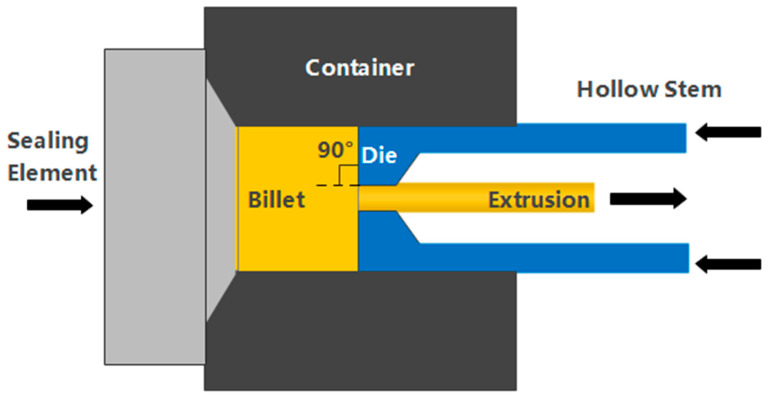
Schematic diagram of the backward-extrusion process.

**Figure 2 materials-16-04588-f002:**
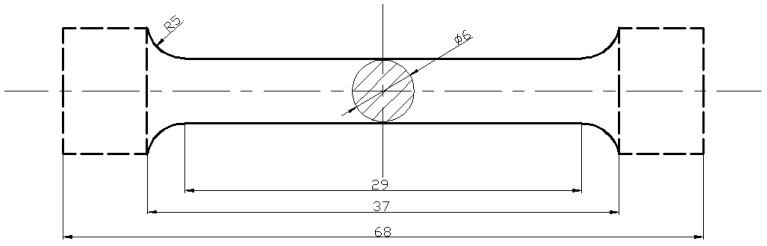
Configuration of the samples used for the tensile tests (unit: mm).

**Figure 3 materials-16-04588-f003:**
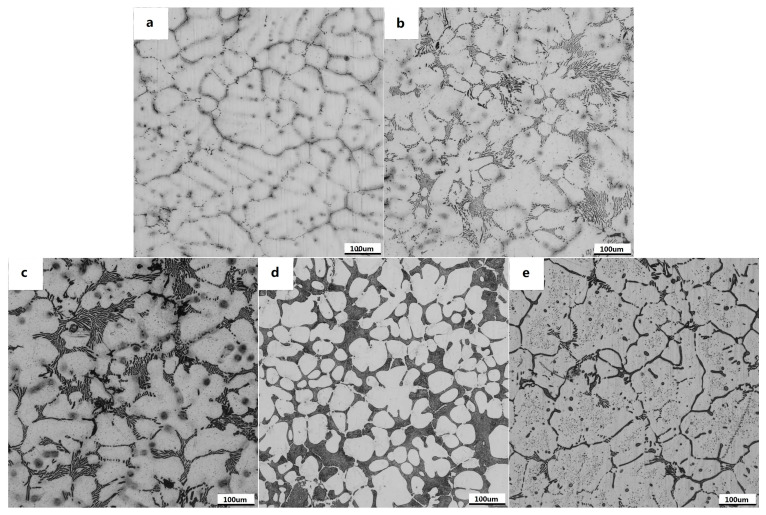
Optical images of the microstructures of the as-cast magnesium alloy: (**a**) Mg-3Sn-1Mn, (**b**) Mg-3Sn-1Mn+0.5La, (**c**) Mg-3Sn-1Mn+1.0La, (**d**) Mg-3Sn-1Mn+1.5La, (**e**) Mg-3Sn-1Mn+2.0La.

**Figure 4 materials-16-04588-f004:**
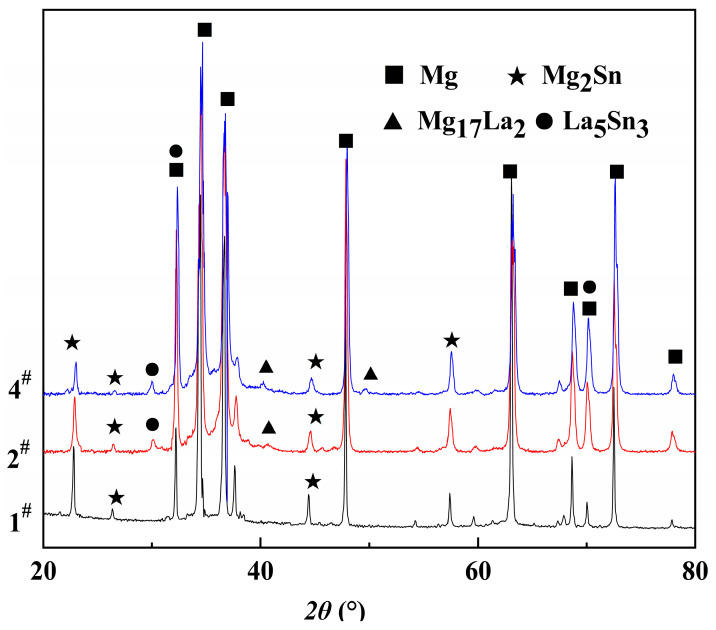
XRD patterns for the Mg-3Sn-1Mn-xLa alloys with 1#, 2#, and 4#.

**Figure 5 materials-16-04588-f005:**
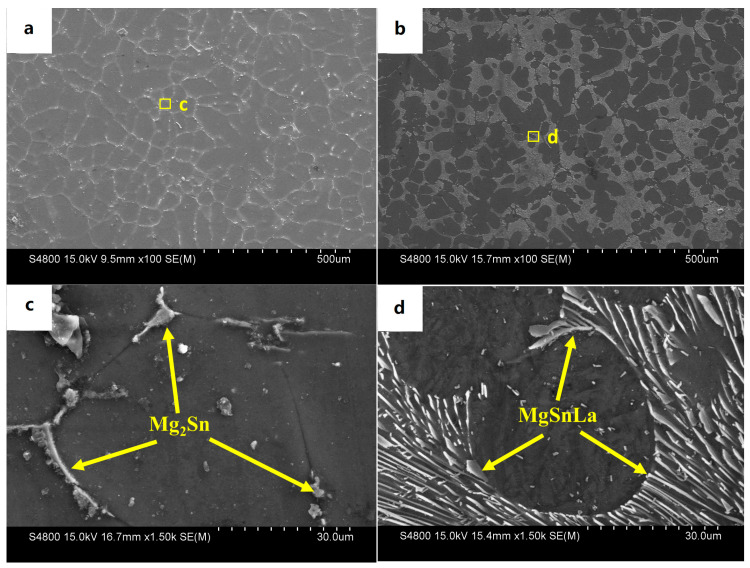
SEM microstructures of as-cast alloys: (**a**,**c**) Mg-3Sn-1Mn, (**b**,**d**) Mg-3Sn-1Mn+1.5La.

**Figure 6 materials-16-04588-f006:**
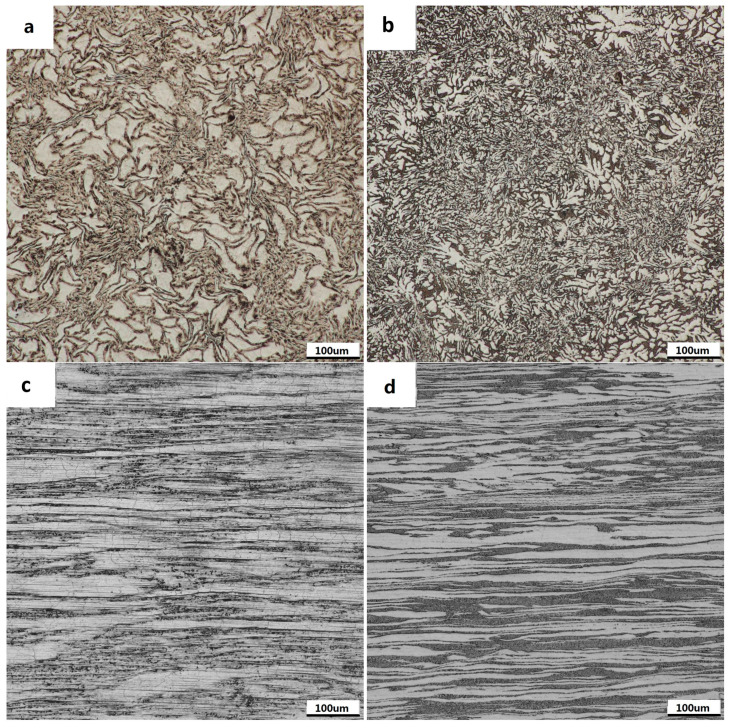
Optical microstructures of Mg-Sn-Mn-xLa alloys backward extruded in transverse and extruded. (**a**,**c**) Mg-3Sn-1Mn (**b**,**d**) Mg-3Sn-1Mn+1.5La.

**Figure 7 materials-16-04588-f007:**
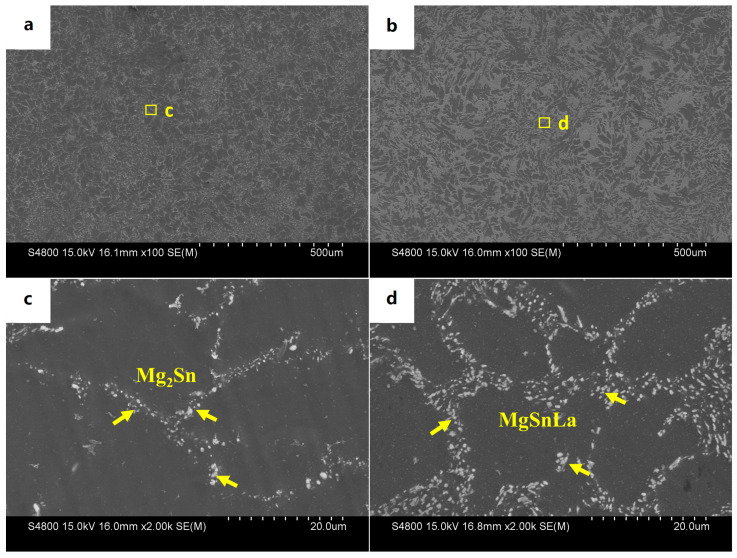
SEM microstructures of Mg-Sn-Mn-xLa backward-extruded alloys.

**Figure 8 materials-16-04588-f008:**
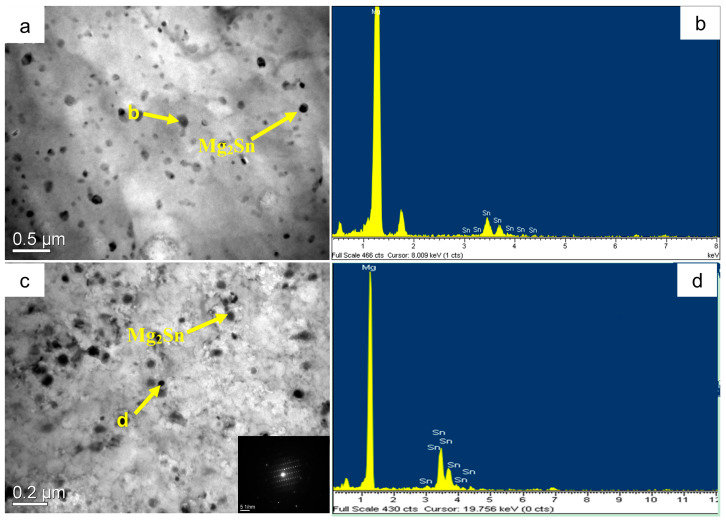
TEM microstructures of Mg-3Sn-1Mn-xLa and composition analysis. (**a**,**b**) Mg-3Sn-1Mn, (**c**,**d**) Mg-3Sn-1Mn-1.5La.

**Figure 9 materials-16-04588-f009:**
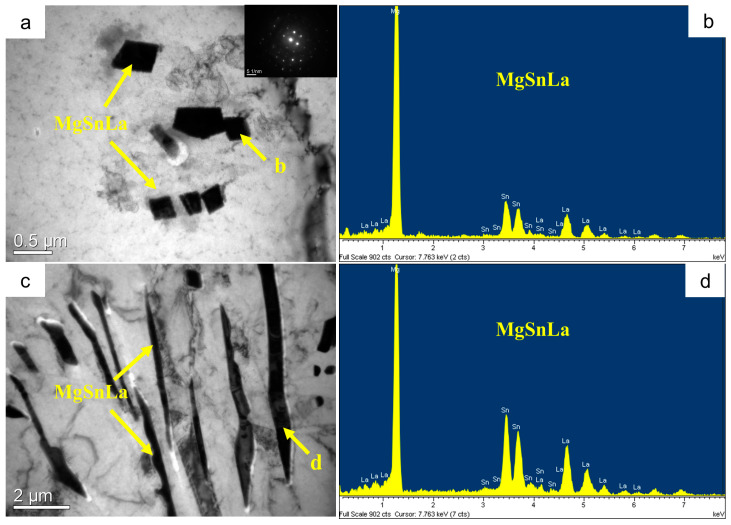
TEM microstructures of Mg-3Sn-1Mn-1.5La and composition analysis.

**Figure 10 materials-16-04588-f010:**
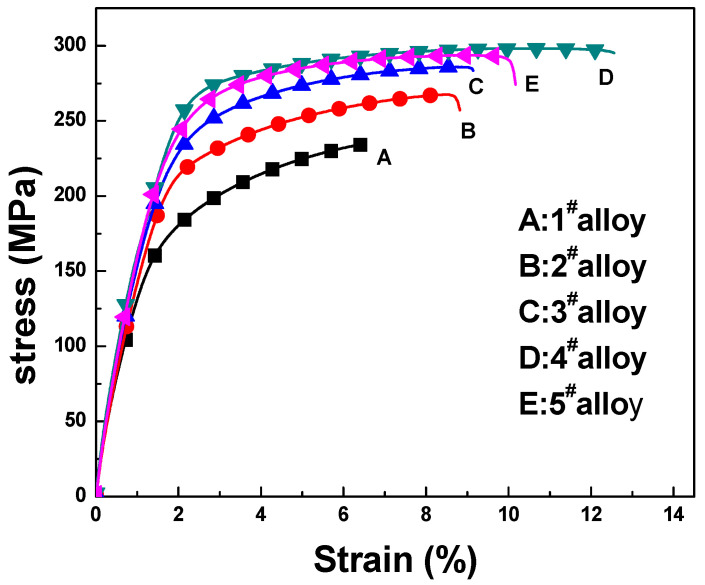
Stress–strain curves of Mg-3Sn-1Mn-xLa alloys with backward extruded.

**Figure 11 materials-16-04588-f011:**
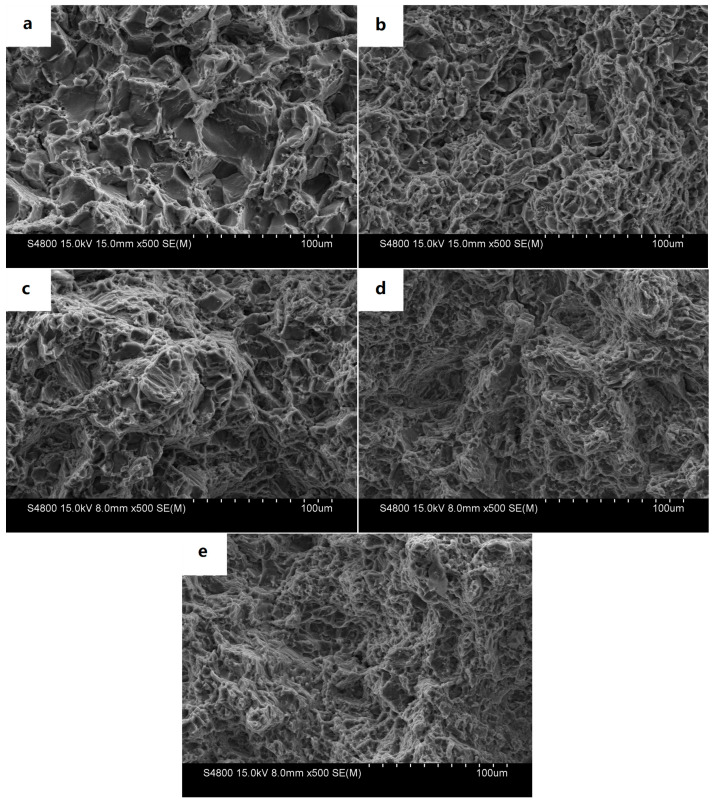
SEM micrographs of the fracture morphologies of the tested specimens: (**a**) Mg-3Sn-1Mn, (**b**) Mg-3Sn-1Mn+0.5La, (**c**) Mg-3Sn-1Mn+1.0La, (**d**) Mg-3Sn-1Mn+1.5La, (**e**) Mg-3Sn-1Mn+2.0La.

**Figure 12 materials-16-04588-f012:**
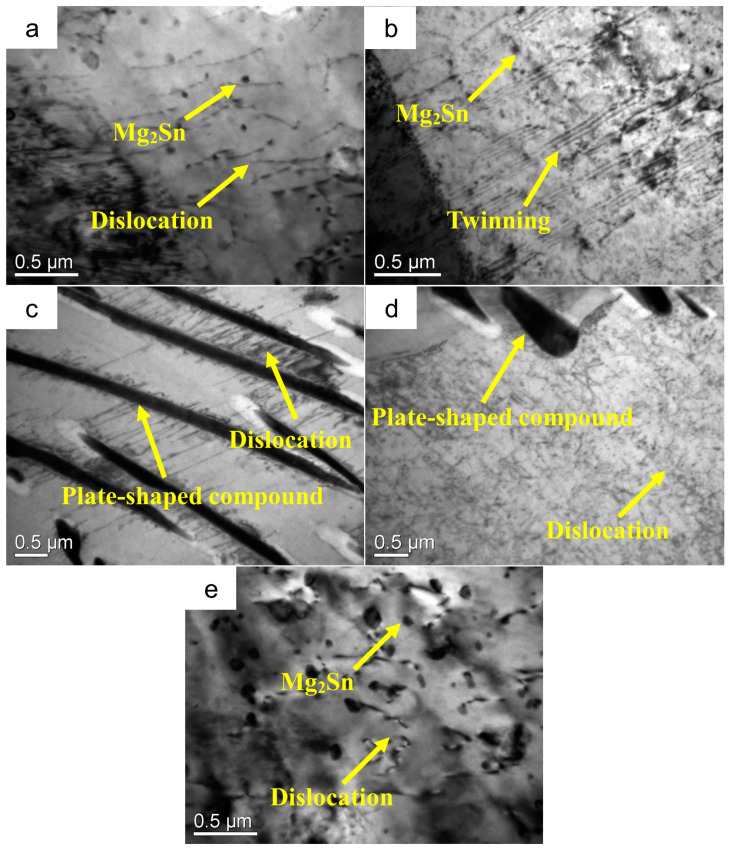
TEM images of backward-extrusion Mg alloy after tensile deformation (**a**,**b**) Mg-3Sn-1Mn, (**c**–**e**) Mg-3Sn-1Mn+1.5La.

**Table 1 materials-16-04588-t001:** Detailed composition of Mg-Sn-Mn-xLa alloys.

Number	Nominal Alloy	Sn (wt.%)	Mn (wt.%)	La (wt.%)	Mg
#1	Mg-3Sn-1Mn	2.81	0.90	0	Bal
#2	Mg-3Sn-1Mn+0.5La	2.79	0.89	0.46	Bal
#3	Mg-3Sn-1Mn+1.0La	2.83	0.91	0.94	Bal
#4	Mg-3Sn-1Mn+1.5La	2.88	0.93	1.45	Bal
#5	Mg-3Sn-1Mn+2.0La	2.85	0.90	1.92	Bal

## Data Availability

The data presented in this study are available on request from the corresponding author.

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
