# Peer review of "Study on Microstructural Evolution and Mechanical Properties of Mg-3Sn-1Mn-xLa Alloy by Backward Extrusion"

_materials, 2023, doi:10.3390/ma16134588_

Round 1

Reviewer 1 Report

The authors presented an article about “Study on microstructural evolution and mechanical properties of Mg-3Sn-1Mn-xLa Alloy by backward extrusion.”

In this study, the authors investigated the microstructure and tensile properties of Mg-3Sn-1Mn-xLa alloys. Although the microstructure investigation studies have been carried out very successfully, only the tensile test has been examined in the mechanical properties section. Failure to test for hardness can be considered a major shortcoming.I think the paper is well organized and appropriate for the “Materials” journal, but the paper will be ready for publication after major revision.

·       The abstract looks good. Please include all significant numerical results.

·       What is the problem? Why was the manuscript written? Please explain the reason in the introduction part. In the last paragraph of the introduction, the novelty of the study and the differences from the past in detail should be expressed.

·       Please add a figure containing the macro photograph of the test instruments and the produced materials to the material method section.

·       Hardness tests have not been performed for the produced materials. This is a huge shortcoming. How do the authors explain this?

·       The article contains only one mechanical property. Please rearrange the article title.

·       Give information about the production of the materials used in the study.

·   Are the mechanical tests carried out in accordance with the test standards? Please specify test standards in methods section.

·       Explain the properties of the phases obtained according to the XRD results. Expand the XRD results. 

·       Please fix the typographical and eventual language problems in the paper.

·       The paper is well-organized, yet there is a reference problem. First, your reference list contains no paper from the “Materials” journal. If your work is convenient for this journal’s context, then there are many references from this journal. Secondly, cited sources should be primary ones. Namely, the indexed area shows the power of a paper and directly your paper’s reliability. Please make regulations in this direction.

*** Authors must consider them properly before submitting the revised manuscript. A point-by-point reply is required when the revised files are submitted.

Please fix the typographical and eventual language problems in the paper.

Reviewer 2 Report

The paper investigates Mg-3Sn-1Mn alloy with the addition of lanthanum. The authors vary the lanthanum content and then extrude the material and study its structure and mechanical properties. It is found that the addition of lanthanum leads to the formation of secondary particles that cause the structure of the cast alloy to refine. After extrusion, these particles inhibit the dislocation motion during deformation and harden the material. The result is quite good, the research is relevant and interesting. There are a few comments:

1.               Table 1: as a rule, samples are labeled #1, not 1#. Here and elsewhere in the text.

2.               Figure 3: In fact, it is not always possible to clearly distinguish from the optical images where the intermetallides are and where the contaminants or eutectics are. At least in the above images it is not possible to clearly identify intermetallides. Definitely they should be indicated. Perhaps it is worth giving more detailed images.

3.               Lines 116 - 134: There is a discussion about the size of the dendrites. From the figures, of course, this is clearly evident, but I suggest that quantitative measurements should be made anyway.

4.               Figures 8 and 9: The given electronograms are not of sufficient size and quality to confirm the results discussion. Also the electronograms are worth deciphering.

5.               Figure 9: still not quite clear your position regarding the particles. With XRD you detected La5Sn3, Mg2Sn, and Mg17La2 particles. Of course, I understand that on SEM images in the interdendritic area you cannot separate them, so you very tentatively call this mixture of particles MgSnLa. However, with TEM phase indication, it's quite different. You have particles of quite definite phases that you can identify with an electronogram and a dark field. Of course, EDS analysis won't help you identify these particles, because the size of the probe is much larger than the particles. So you're probing the alloy matrix and you're finding lanthanum and tin there. So in this case it's not really correct to call these particles MgSnLa. You should identify them by diffraction.

6.               Line 264: Sn3RE5. Actually, based on the work context, I can guess what it means, but still, the abbreviation should be explained.

7.               An unnecessary note: it would be very interesting to see the results of mechanical tests of the produced alloys before extrusion.

Round 2

Reviewer 1 Report

Thank you for reply

Minor editing of English language required